# Conformational Sampling Deciphers the Chameleonic Properties of a VHL-Based Degrader

**DOI:** 10.3390/pharmaceutics15010272

**Published:** 2023-01-12

**Authors:** Giuseppe Ermondi, Diego Garcia Jimenez, Matteo Rossi Sebastiano, Jan Kihlberg, Giulia Caron

**Affiliations:** 1Department of Molecular Biotechnology and Health Sciences, University of Torino, Quarello 15, 10135 Torino, Italy; 2Department of Chemistry—BMC, Uppsala University, SE-75123 Uppsala, Sweden

**Keywords:** bRo5, conformational sampling, early drug discovery, molecular chameleon, NMR, PROTAC

## Abstract

Chameleonicity (the capacity of a molecule to adapt its conformations to the environment) may help to identify orally bioavailable drugs in the beyond-Rule-of-5 chemical space. Computational methods to predict the chameleonic behaviour of degraders have not yet been reported and the identification of molecular chameleons still relies on experimental evidence. Therefore, there is a need to tune predictions with experimental data. Here, we employ PROTAC-1 (a passively cell-permeable degrader), for which NMR and physicochemical data prove the chameleonic behaviour, to benchmark the capacity of two conformational sampling algorithms and selection schemes. To characterize the conformational ensembles in both polar and nonpolar environments, we compute three molecular properties proven to be essential for cell permeability: conformer shape (radius of gyration), polarity (3D PSA), and the number of intramolecular hydrogen bonds. Energetic criteria were also considered. Infographics monitored the simultaneous variation of those properties in computed and NMR conformers. Overall, we provide key points for tuning conformational sampling tools to reproduce PROTAC-1 chameleonicity according to NMR evidence. This study is expected to improve the design of PROTAC drugs and the development of computational sustainable strategies to exploit the potential of new modalities in drug discovery.

## 1. Introduction

The principles of the 3Rs (*Replace* animal experiments with alternative methods, when possible, *Refine* them so that pain and discomfort are avoided, and *Reduce* the number of employed animals) are commonly seen as a framework for conducting high-quality science with a higher focus on developing alternative approaches that avoid the use of animal models. The 3Rs framework is particularly relevant when new modalities [1,2,3] such as targeted protein degradation (TPD) are considered, since a definitive set of rules to guide their design and optimization has not yet been defined. TPD involves recruiting a protein of interest (POI) to a ubiquitin E3-ligase with heterobifunctional molecules (generally referred to as PROteolyis TArgeting Chimeras, PROTACs) that contain binding motifs for each partner coupled via a linker [4]. Most PROTACs do not respect Lipinski’s Rule of 5 (Ro5) [5], but reside in the so-called beyond-the-Rule-of-5 (bRo5) chemical space [1,3,6,7]. As the Ro5 defines the chemical space in which it is more likely to find small molecules with favourable pharmacokinetic properties, it is reasonable to speculate that bRo5 molecules, despite good pharmacodynamic features, have a higher risk of displaying a poor ADME (adsorption, distribution, metabolism, and excretion) profile [8,9]. This mainly originates from the counter position between two properties: solubility and passive membrane permeability. Indeed, molecules above a certain molecular weight have a risk of being either too polar, resulting in adequate solubility and poor membrane permeability, or the opposite, if too lipophilic [10,11,12,13]. This is verified for some bRo5 failed drug candidates, but it is widely known that many successful drugs lie in the bRo5 chemical space [8,12,14,15].

One explanation for the fact that orally bioavailable drugs can exist in the bRo5 space lies in the concept of chameleonicity [16,17], namely the ability of a molecule to adapt to environments with different properties [13,18,19,20]. This bRo5-specific feature allows molecular chameleons to be sufficiently polar as to be soluble in an aqueous environment and interact with the receptor, while also being lipophilic enough to be cell-permeable [21]. Intuitively, chameleonicity is linked to the capacity of a molecule to be flexible [22,23], and thereby to populate a large conformational space. Although a certain degree of flexibility is necessary, it is not sufficient; dynamic intramolecular interactions (mainly intramolecular hydrogen bonds, IMHBs) and hydrophobic collapse must act as drivers for the conformational changes and the resulting property changes [18,24,25,26,27,28]. Further than a mere hypothesis, the existence of molecular chameleons has been experimentally proven by us and other colleagues [13,19,20,29]. Therefore, bRo5 and in particular PROTAC drug discovery needs strategies to design new molecules that behave as molecular chameleons.

A few experimental and computational techniques have recently been employed to monitor chameleonic behaviour. For instance, chromatographic indexes such as ChamelogD [10] and the variation of the capacity factor in a nonpolar chromatographic system (PLRP-S) can be informative in this respect [30,31]. However, the experimental technique of choice is Nuclear Magnetic Resonance (NMR) spectroscopy, which can obtain the atomic resolution and dynamic details of a molecule in solution [32,33,34,35] and monitor variation in physicochemical properties [20]. Concerning computational predictions, recent results indicate that using different tools to generate conformational ensembles provides different sets of conformers [30]. Thus, there is a strong need to benchmark in silico predictions with experimental data. 

PROTAC-1 [19] (Figure 1), a cell-permeable (−log P_e_ is 5.85 cm/s in PAMPA), anticancer degrader [36,37] that resides in the bRo5 space [1] (see Appendix A), is a good model to explore how computational tools can describe conformational variability experimentally determined in polar and nonpolar media. There are two main reasons for this: (a) a previous thorough NMR characterization revealed that PROTAC-1 is a molecular chameleon, with a high proportion of folded conformers in nonpolar environments and, conversely, more linear ones in water [19]; and (b) PROTAC-1 is predicted to be neutral at pH 7.4 and in most of the pH range (Appendix A), simplifying the calculations.

For these two reasons, we employ PROTAC-1 to benchmark the capacity of a few conformational sampling (CS) protocols and selection schemes to reproduce the results from the NMR-based solutions’ conformational analysis and physicochemical data. To this end, we provide an analysis of the experimental solution conformers and compare them with computed conformational ensembles. We monitor molecular descriptors rather than comparing conformers through the root-mean-square deviation of atomic positions (RMSD), as it has proved to be more relevant for bRo5 molecules [20]. Specifically, we focus on three molecular descriptors: (a) the surface-accessible 3D polar surface area (3D PSA), (b) the radius of gyration (R_gyr_) [9,20], and (c) the number of intramolecular hydrogen bonds (nIMHB) [26]. Describing polarity via 3D PSA is a valid strategy to explain important properties such as permeability and solubility [38,39,40,41]. Notably, the 3D PSA method employed here is directly comparable with topological PSA (TPSA) values [30]. R_gyr_ is a descriptor mainly related to the shape of a conformer proving useful to model the cell permeability of flexible molecules [9,20], and nIMHB is a widely known driver of chameleonicity [24,26].

## 2. Materials and Methods

Initial 3D geometry was generated with Maestro (Schrödinger Release 2022-3: Maestro, Schrödinger, LLC, New York, NY, USA, 2021) carefully checking all chiral centres. 

The conformational samplings were performed using the different algorithms available in Maestro, in particular mixed torsional/low-mode sampling (LMOD) and Monte-Carlo torsional sampling (MMCM) [42]. The force field OPLS3e [43] (with default parameters) was employed using an implicit solvent treatment for water and chloroform as implemented in Maestro.

The energy-weighted plot was obtained using DataWarrior (http://www.openmolecules.org/datawarrior/ (accessed on 20 December 2022), version 5.2.1, 2021). The difference between the energy of each conformer and the MEC energy was calculated and then converted into a probability using all the conformers according to the Boltzmann equation. These probabilities were used to set the marker size to strictly proportional mode and then manually reduced to a quarter in the DataWarrior ruler.

The clustering of the conformers was performed using the default tool available in Vega (version 3.2.2.21, https://www.ddl.unimi.it/ (accessed on 20 December 2022)).

The input for MOPAC2016 was prepared exporting the CS results as mol2 files and then converting such files in MOPAC2016 format using OpenBabel (Open Babel, v.3.1.1; http://openbabel.org (accessed on 20 December 2022)) [44]. The MOPAC2016 keywords were added using OpenBabel options. The structures minimized with MOPAC2016 were converted to mol2 using OpenBabel again.

A selection of conformations was minimized with MOPAC2016 (version 21.237 W, Stewart Computational Chemistry, https://openmopac.net/ (accessed on 20 December 2022)). To mimic a water environment (dielectric constant ε = 80) and the interior of the membrane (dielectric constant ε = 4.81), minimization was performed applying the Conductor-like Screening Model (COSMO) 97 continuum approach implemented in MOPAC2016.

The calculation of 3D PSA and R_gyr_ was performed in VEGA ZZ [45] (http://www.vegazz.net/ (accessed on 20 December 2022)) by importing the conformer structures as unique files in mol2 format. All the descriptors from Vega ZZ were calculated with standard settings. Specifically, 3D PSA had a probe radius with the default value (0); thus, it should be regarded as effective molecular PSA.

Intramolecular hydrogen bonds, IMHBs, were counted with USCFChimera 1.16 [46] (https://www.rbvi.ucsf.edu/chimera/ (accessed on 20 December 2022)). Hydrogens bound to nitrogen, oxygen, and sulphur were considered as donors. Nitrogen, oxygen, and sulphur atoms with lone pairs were considered as acceptors. The parameters 0.4 Å bond distance and 20° (HBD-HBA angle) were used [47]. These parameters correspond to the USCFChimera default settings.

Microsoft Excel version 2010 (www.microsoft.com (accessed on 20 December 2022)) was employed for data analysis. OSIRIS DataWarrior was used to generate the infographics.

## 3. Results and Discussion

### 3.1. Analysis of NMR Data

The solution conformations of PROTAC-1 were reported in a previous publication [19]. Time-averaged NMR data, i.e., nuclear Overhauser effects (NOEs), were deconvoluted into individual conformers by employing the algorithm-termed NMR analysis of flexibility in solution (NAMFIS) [48]. In brief, experimental proton–proton distances, determined by NOESY spectroscopy, and dihedral angles, calculated from ^3^*J*_H,H_ couplings (when available), represent the input for NAMFIS. In parallel, NAMFIS requires a theoretical conformational ensemble providing a comprehensive coverage of the molecule’s conformational space. The algorithm strives to find the best fit of the interproton distances and dihedral angles determined by NMR spectroscopy to back-calculated data from a probability-weighted combination of conformations selected from the theoretical ensemble. With this method, the NMR solution structures of PROTAC-1 were determined in three different solvents.

In the present study, we focus on the conformers obtained in two solvents [19]: chloroform, mimicking the interior of the cell membranes, and a water/DMSO mixture (from here on named water for simplicity), mimicking the extracellular environment. Eight and ten conformations were found in chloroform and water, respectively. In chloroform, the conformations are either highly folded (90%) or semi-folded (10%). In contrast, the ensemble in water is composed of a mixture of highly folded (29%), semi-folded (56%), and linear (15%) conformations [19]. For all the NMR conformers, we calculated the three descriptors successfully used in previous publications [13,19,20,30] to characterize the chameleonic properties of bRo5 compounds: the solvent-accessible 3D polar surface area (3D PSA), the radius of gyration (R_gyr_), and the number of intramolecular hydrogen bonds (nIMHB). 

As recently shown for other bRo5 molecules [30], the simultaneous monitoring of multiple descriptors proved effective in obtaining information on the possible chameleon-like behaviour. Figure 2 shows results for PROTAC-1. In Figure 2A, the size of each point reflects the abundance derived from the NMR data. In Figure 2B, a similar representation is color-coded by the number of IMHBs.

The combination of 3D PSA and R_gyr_ highlights a solvent-induced chameleonic behaviour of PROTAC-1, with more spherical and less polar conformations in chloroform than in water (Figure 2A). When the IMHBs count is introduced, no trend is found between solvents, but conformations with lower 3D PSA and R_gyr_ in each solvent have two IMHBs instead of one (Figure 2B). These results suggest that the absolute number of IMHBs does not clearly influence the variation in 3D PSA and R_gyr_ between solvents. Consideration of the conformer population provides additional information (Figure 2A,B): the most populated conformations in water display high 3D PSA and two R_gyr_ classes, while the chloroform conformations displaying “abnormally” high 3D PSA tend to be less populated. Finally, no conformer reaches the TPSA value of PROTAC-1 (265 Å^2^), suggesting that the molecule is not prone to fully exposing its polarity [19]. Even the linear conformation in water (15%) displays one IMHB that reduces the solvent exposure of the involved hydrogen bond donor and acceptor.

Finally, we obtained a rough estimation of the energy range covered by the solution conformers by calculating for each conformer a single point energy with Schrodinger (Sc) [43] using the same force field (OPLS3e with generalized-born/surface-area (GB/SA) solvation model) employed for conformational sampling. The results suggest that the conformer energies are spread over a range of about 16 kcal/mol and 8 kcal/mol in water and chloroform, respectively. The large ranges suggest that fixing an energy threshold in a conformational search is an extremely difficult task as previously found for a set of macrocyclic and non-macrocyclic drugs in bRo5 space [20]. Instead, a valid strategy might be to consider the whole ensemble and explore which are the values of the physicochemical descriptors accessible to the molecule. This idea is reinforced by considering that the molecule can dynamically assume conformations calculated to have high energy when crossing cell membranes, or when moving between two immiscible solvents due to interactions with the environment.

### 3.2. Experimental Physicochemical Data to Monitor the Chameleonicity of PROTAC-1

As previously reported [30,31] and mentioned in the Introduction, the propensity of compounds to behave as molecular chameleons can be monitored with the variation in the capacity factor in a nonpolar polymeric chromatographic system, (PLRP-S) [30,31]. We measured the retention of PROTAC-1 with a PLRP-S column by changing the composition of the mobile phase (ranging between 50 and 100% ACN, see Figure 3). The reverse-phase nature of the stationary phase causes the retention time (and thus the logarithm of the capacity factor log k′) of lipophilic molecules to linearly decrease for mobile phases with progressively higher acetonitrile fractions. In practice, the higher the amount of ACN, the lower the log k′ PLRP-S. Figure 3 shows that toluene (grey triangles, the control) behaves in this way. Notably, for PROTAC-1, the log k′ PLRP-S at 100% ACN is much higher than its expected value (Figure 3, green dots). This experimental finding is interpreted by the propensity of PROTAC-1 to display conformations with lower polarity in nonpolar environments, i.e., to behave as a molecular chameleon.

Overall, the chromatographic results confirm that PROTAC-1 behaves as a molecular chameleon, without providing any additional information about the nature of the intramolecular interactions responsible for this behaviour.

### 3.3. Computational Analysis

After the analysis of experimental data, we focused on in silico approaches. We again reiterate the importance of having reliable in silico methods to promote a sustainable drug discovery process, reducing the environmental impact and the use of experimental animal models, a particularly pressing issue in new research fields such as that of developing new modalities.

#### 3.3.1. Conformational Analysis Strategies

Performing a conformational analysis with the aim of reproducing experimental data is not a trivial task. Below, we briefly discuss the factors that should be considered to set up a sound strategy. 

First, the conformational space should be sampled as completely as possible. To tackle this point, the number of generated conformers should be increased and/or a suited algorithm intrinsically exploring a wider portion of the conformational space should be used [20]. In addition, it is often needed to access higher energy conformers by considering a wider energy range. Selecting a valid energy range is a more complex problem than choosing a higher number of selected conformers, essentially because the energy window to be selected is often dependent on the complexity of the conformational landscape that each molecule can express and by the selected force field. Another issue of conformational analysis concerns which conformers should be selected as the most representative of the whole ensemble. Choosing the minimum energy conformer (MEC) is a dangerous strategy, as previously reported [20]. Therefore, conformational sampling is difficult to specifically tailor to chameleonicity. This latter in fact requires that simulations are performed in two different environments. Since CS protocols adopt an implicit solvent model (faster but less accurate than an explicit approach), this represents another source of bias. A final remark is related to the NAMFIS procedure for which it is essential that a theoretical ensemble that covers conformational space as completely as possible is used. For PROTAC-1, this was ensured by Monte Carlo conformational analysis using five different force fields, each with the GB/SA solvation models for chloroform and water in the Schrödinger software [19]. These 10 ensembles were then combined into one, and redundant conformations were eliminated to give a diverse ensemble that was used for the deconvolution of the experimental NOE data both for chloroform and water. Since the study only focused on the possibility of reproducing the chameleon-like behaviour detected by NMR in the two solvents, we used the same software as in the NAMFIS analysis. In addition, we preferred a simplified procedure for each solvent based only on one of the force fields used for the NAMFIS analysis, which is more suitable for application to a larger number of compounds. At the initial stage of drug discovery, this is a more realistic situation, possibly reserving more in-depth analysis for when a smaller number of compounds are prioritized.

Based on the aforementioned considerations, we performed a conformational sampling with the Schrodinger (Sc) [42] software using different algorithms and parameters (Table 1), then evaluated to what extent these computational approaches can reproduce the NMR results in terms of physicochemical descriptors (3D PSA, R_gyr_, and nIMHB), as pointed out in previous papers [19,20,30]. We first checked whether the standard protocol (Sc01) suggested by the manufacturers could reproduce the experimental data, then we increased the number of generated conformers. Altogether, we used two search algorithms: the default mixed torsional/low-mode sampling (Sc01–Sc03) and the Monte Carlo torsional sampling (Sc04–Sc06), with the OPLS3e [49] force field. Finally, to explore potential bias from the force field, we clustered the conformers and performed a minimization of a representative conformer of each cluster with a semi-empirical, quantum mechanics protocol by using the software MOPAC (named post-CS minimization) [50].

#### 3.3.2. Results of the Conformational Sampling

All of the results are provided in Appendix A. The first methodological insight regards the number of conformations to be explored by the algorithm and those retained at the end of the sampling. Interestingly, it seems that, regardless of the algorithm used, generating 10^4^ and 10^5^ conformations returns the same conformational ensembles as shown by the comparison of Sc02 vs. Sc03 and Sc05 vs. Sc06 (Appendix A).

The next step consisted of analysing the 3D PSA vs. R_gyr_ plots (Figure 4). Three different selection schemes were applied to the conformers: (a) all conformations are considered, (b) conformers within an energy window of less than 3 kcal/mol from the MEC were selected, and (c) the conformations were energy-weighted. The ensembles from runs Sc03 and Sc06 were not analysed since they were identical to those from runs Sc02 and Sc05, respectively. All the adopted selection schemes suggest a decrease in 3D PSA when moving from water to chloroform. This is an expected result that suggests that PROTAC-1 decreases its 3D PSA moving from a polar to a non-polar environment. There is an overlap in 3D PSA in the two solvents, better highlighted using all conformations or a weighted scheme (Figure 4). This overlap suggests that some conformations exploiting similar physicochemical properties might be more prone to switch from one solvent to another [51]. 

Overall, the energy-weighted protocol Sc05 provides the best reproduction of the chameleonic behaviour of PROTAC-1 observed by solution NMR spectroscopy. On the other hand, the default parameters, Sc01, can catch the decrease in 3D PSA (and catch the chloroform conformer with lowest polarity) with a sensible gain in terms of computational time. This result suggests that the conformational sampling procedure must be carefully tuned in terms of the algorithm, the number of generated conformers, and the conformer analysis. In addition, the default algorithm, Sc01, can be used for an estimation of chameleonic behaviour in series of compounds, while the more CPU-expensive protocol Sc05 can be used to gain more insight for uncertain cases. Issues related to R_gyr_ interpretation are discussed below.

Sc05, coupled with the energy-weighted scheme, suggests some other relevant points when CS conformers are compared to the solution ensembles determined by NMR spectroscopy (Figure 5): (a) the CS predicts a higher number of highly populated conformations than NMR, and (b) the range of R_gyr_ and 3D PSA explored by the experimental ensembles is not completely superposable with the CS results. The higher R_gyr_ values of some of the aqueous NMR conformations are especially not reached by CS conformers. This could be explained by the fact that NMR experiments were performed in DMSO–water, while water alone was used for the CS experiments. This is supported by the fact that R_gyr_ is reduced in the NMR study when 10% water is added to DMSO [19], i.e., that PROTAC-1 folds in a different way to expose polar groups when water is added as compared to pure DMSO. One can therefore speculate that the addition of even more water (as in the CS, performed in 100% water) would result in even more folding. We are aware that CS protocols simulating 100% water are not perfectly simulating the experimental NMR conditions, but we chose 100% water for the following reasons: (1) sampling in DMSO–water mixtures would have resulted in a poor solvent simulation by setting a dielectric constant, without the extra parameters optimized for water, and (2) by mimicking a water environment, we aimed to maintain physiological conditions as much as possible.

Finally, we inspected in more detail the conformers with populations predicted to be >4% from the Sc05 protocol (Figure 6). The first observation is that no linear conformer is found, justifying the lack of superposition with high-R_gyr_ conformations from the NMR analysis. Second, in agreement with the NMR analysis, IMHBs are present in both solvents (see Appendix A for further IMHBs pattern details).

#### 3.3.3. Post Minimization with MOPAC

To reduce the bias introduced by any force field, we performed a clustering of the ensemble Sc01. We then took the representative conformers and minimized them with the semi-empirical method PM7 [50] using the quantum mechanics engine MOPAC2016 [50] (see Methods), with an implicit treatment of the solvent. Altogether, two runs were carried out to obtain two sets of conformers for Sc01, one for each solvent. Sc01 is called MO01 after minimization. The results are summarized in the SI (Appendix A). Notably, when comparing the MO01 sets with the NMR data (Figure 7), it is evident that the individuated property spaces are not comparable. Thus, QM-based energy minimization of the conformations using an implicit solvation model resulted in a deterioration of the predictions of the chameleonic behaviour of PROTAC-1.

## 4. Conclusions

Modern drug discovery calls for a refinement of computational strategies apt to fulfil the gap in experimental characterization. New therapeutic modalities such as PROTACS are among the categories that can benefit the most from methods predicting molecular properties such as chameleonicity with a pure in silico approach. 

In this work, we focus on the degrader named PROTAC-1, proven to be a molecular chameleon through state-of-the-art solution NMR studies and physiochemical data. In particular, we explore the capability of in silico protocols to reproduce the solution conformations in polar and nonpolar media, diagnostic for identifying the chameleonic behaviour of the compound.

Overall, our results support that conformational sampling can reproduce NMR solution conformers, although computational settings should be checked according to the information we wish to obtain. To evaluate the 3D PSA variation in the two environments due to chameleonicity effects, default protocols are sufficient and fast. However, to obtain a more accurate reproduction of NMR conformers, the modulation of the algorithm, the number of conformers, and energetic considerations should be introduced. An additional remark concerns the characterization of the R_gyr_ range in the polar environment since experimental conditions, including the presence of DMSO in the NMR solution, cannot be computationally reproduced. 

Overall, this study describes an approach that may be transferable to other PROTACs and bRo5 compounds with the potential to reduce the use of chemical matter, thus promoting sustainability.

## Figures and Tables

**Figure 1 pharmaceutics-15-00272-f001:**
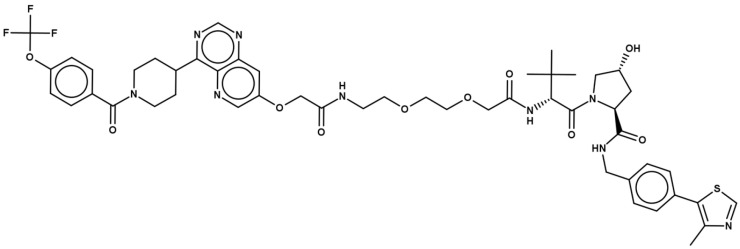
Chemical structure of PROTAC-1.

**Figure 2 pharmaceutics-15-00272-f002:**
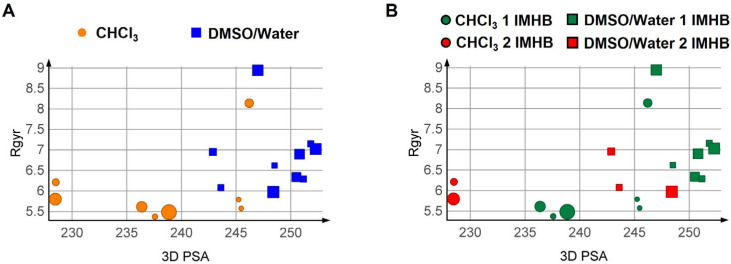
R_gyr_ plotted vs. 3D PSA for the NMR-derived solution conformations of PROTAC-1. (**A**) Colour and shape by solvent, (**B**) colouring by nIMHBs. The marker shape indicates the solvent and the size of the relative population in %.

**Figure 3 pharmaceutics-15-00272-f003:**
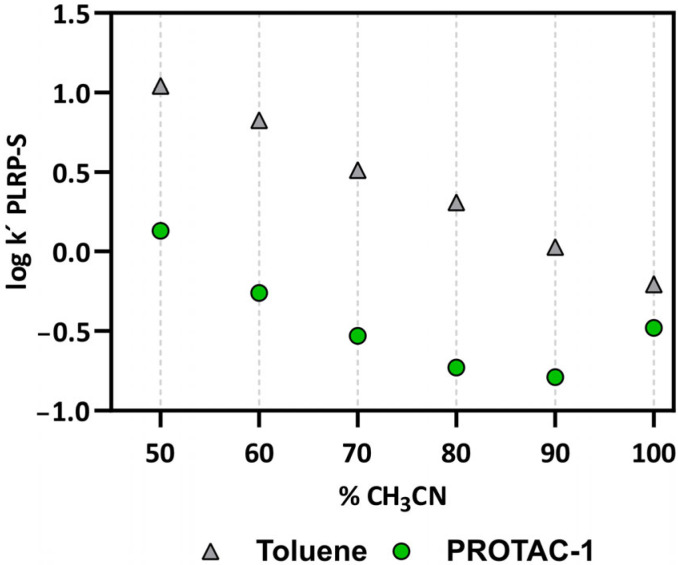
log k′ measured using a PLRP-S stationary phase and increasing percentages of acetonitrile in the mobile phase for PROTAC-1 (green) and toluene (negative control, grey).

**Figure 4 pharmaceutics-15-00272-f004:**
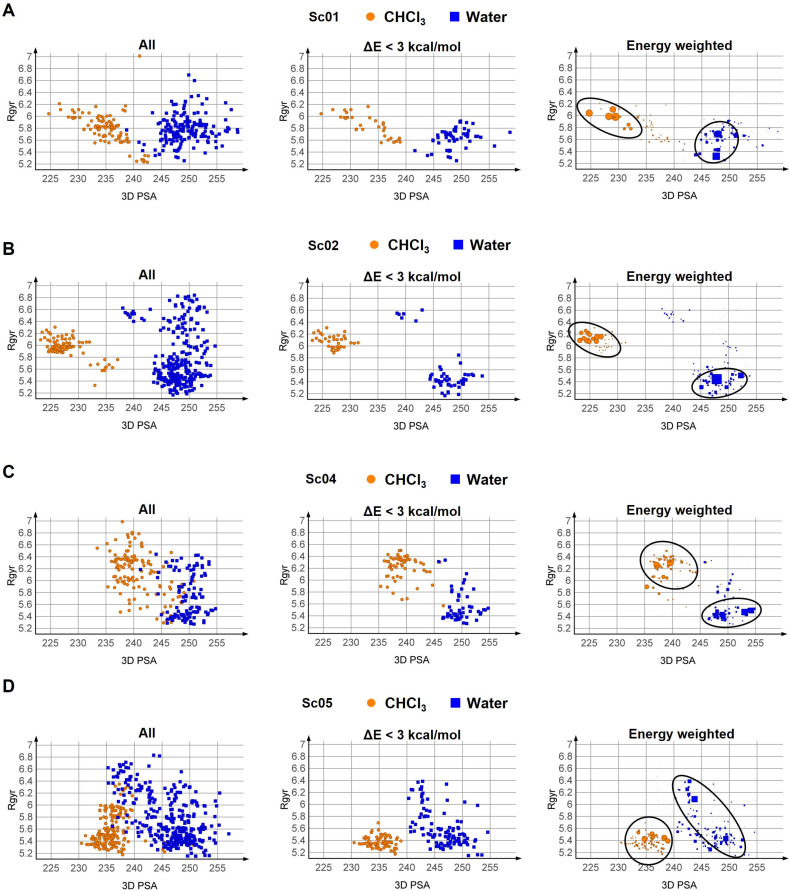
Plots of R_gyr_ vs. 3D PSA for the ensembles obtained with the different protocols. Three protocols for conformational sampling were used in combination with three schemes for selecting the conformations. (**A**) Sc01, (**B**) Sc02, (**C**) Sc04, and (**D**) Sc05. Black circles highlight high-probability conformer groups. Sc03 and Sc06 are not reported since they provided identical ensembles to Sc02 and Sc05, respectively.

**Figure 5 pharmaceutics-15-00272-f005:**
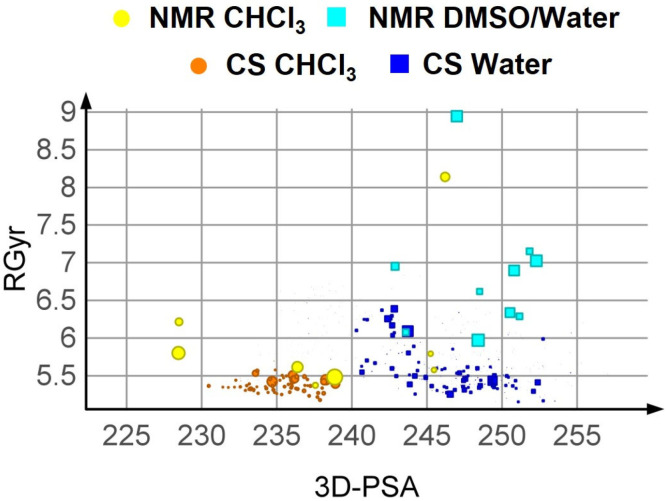
Comparison between CS (Sc05, weighted scheme) and the NMR results.

**Figure 6 pharmaceutics-15-00272-f006:**
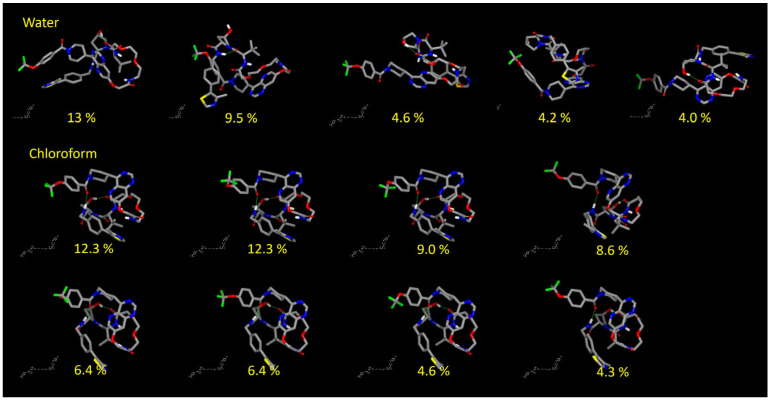
Most relevant conformations, probability greater than 4%.

**Figure 7 pharmaceutics-15-00272-f007:**
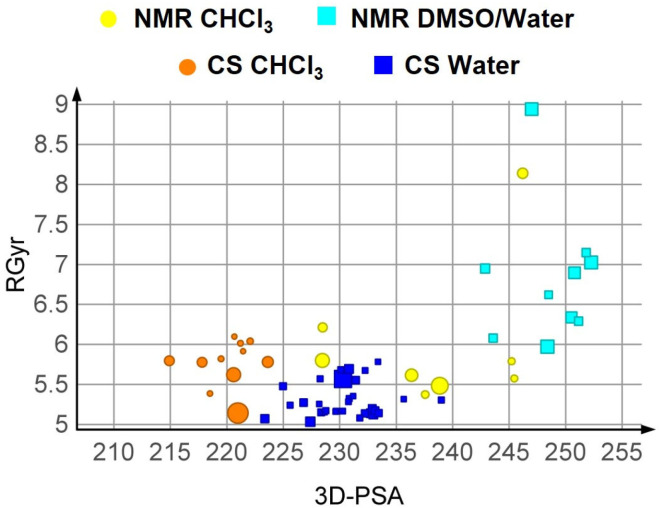
Comparison between NMR and MOPAC post-minimization results (MO1), weighted plot.

**Table 1 pharmaceutics-15-00272-t001:** Detailed description of the protocols used for conformational sampling.

CS Protocols				
**Software**	**Protocol Label**	**Algorithm**	**Parameters**	**FF; Solvation**
Schrodinger	Sc01	Mixed torsional/low-mode sampling. LMOD	max number steps = 10^3^	OPLS3e; Generalized-Born/Surface-Area (GB/SA) model
Schrodinger	Sc02	Mixed torsional/low-mode sampling. LMOD	max number steps = 10^4^	OPLS3e; Generalized-Born/Surface-Area (GB/SA) model
Schrodinger	Sc03	Mixed torsional/low-mode sampling. LMOD	max number steps = 10^5^	OPLS3e; Generalized-Born/Surface-Area (GB/SA) model
Schrodinger	Sc04	Torsional sampling (MCMM)	max number steps = 10^3^	OPLS3e; Generalized-Born/Surface-Area (GB/SA) model
Schrodinger	Sc05	Torsional sampling (MCMM)	max number steps = 10^4^	OPLS3e; Generalized-Born/Surface-Area (GB/SA) model
Schrodinger	Sc06	Torsional sampling (MCMM)	max number steps = 10^5^	OPLS3e; Generalized-Born/Surface-Area (GB/SA) model
**Post-CS Minimization**				
**Software**	**Method Label**	**Starting Conformations**	**Parameters**	**Hamiltonian; Solvation**
MOPAC	MO1	Clusters from Sc01	Default	PM7; Conductor-like Screening Model

## Data Availability

The data presented in this study are available on request from the corresponding author.

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
