# Peer review of "Conformational Sampling Deciphers the Chameleonic Properties of a VHL-Based Degrader"

_pharmaceutics, 2023, doi:10.3390/pharmaceutics15010272_

Round 1
Reviewer 1 Report
December 26, 2022
Title: Conformational sampling deciphers the chameleonic properties of a VHL based degrader
Authors: Giuseppe Ermondi, Diego Garcia Jimenez, Matteo Rossi Sebastiano, Jan Kihlberg, and Giulia Caron
Overview and general recommendation:
In this manuscript, “chameleonicic” conformational property of a passive cell-permeability degrading agent (PROTAC-1) in two different solvent environments were computationally examined. They examined different methods and parameter sets, and propose two methods, CPU effective LMOD and CPU expensive MCMM, which reasonably reproduce the chameleonicity observed in NMR (their previous study, Ref. 19). The analysis is well done and the manuscript is well written. I recommend this work for the publication in Parmaceutics, but suggest some minor revisions.
Minor comments:
(1) This work relies on their precedent works, Refs. 19, 20, and 34. The NMR data appear to have been taken from one of them (Ref. 19), but the original data (e.g. 90 % folded and 10 % semi-folded populations in Page 4) are difficult to find. In addition, Ref. 34 also reports the computational results on PROTAC. Please clearly indicate which is the original of this work, and which is from the previous work.
(2) The overlap of 3D PSA between two solvents (Fig. 4) is evident only when using all conformations and not in the energy weighted graph. The conformations in this region share the similar 3D PSA but exhibit different Rg. What about hydrogen bonds (IMHBs) ? I wonder if the notion like “congruent state” is applicable.
(3) The procedure for obtaining “energy weighted” ensemble should be described in detail so that it can be reproduced.
(4) I wonder if molecular dipole moments differ among representative conformers.
(5) The term “conformer selection” can be confused with the classical “conformer selection vs induced fit”.

Author Response
Minor comments:
- This work relies on their precedent works, Refs. 19, 20, and 34. The NMR data appear to have been taken from one of them (Ref. 19), but the original data (e.g. 90 % folded and 10 % semi-folded populations in Page 4) are difficult to find. In addition, Ref. 34 also reports the computational results on PROTAC. Please clearly indicate which is the original of this work, and which is from the previous work.
The reviewer is right. Indeed, it was not fully clear which data are original and which should be referenced. In brief, citations 20 and 34 are just methodological, while ref 19 provides the experimental conformations of PROTAC-1 determined with NMR and used as benchmark for the computational methods. We added one sentence in the introduction to make it clearer. In practice, in section 3.1 we took experimental NMR conformers reported in [19] and applied a simultaneous monitoring of 3D PSA, Rgyr and nIMHB in a novel manner. We changed the text and the section title (to “Analysis of NMR Data”) accordingly to improve the understandability.
(2) The overlap of 3D PSA between two solvents (Fig. 4) is evident only when using all conformations and not in the energy weighted graph. The conformations in this region share the similar 3D PSA but exhibit different Rg. What about hydrogen bonds (IMHBs) ? I wonder if the notion like “congruent state” is applicable.
Good point. In the energy weighted diagram, the absence of overlapping is due to the fact that this latter involves high-energy conformers. In both 3D PSA and Rgyr overlapping regions when present (e.g., see Figure 4D) some conformations besides sharing 3D PSA and Rgyr values, also share the same nIMHBs, data not shown. However, we agree with the referee that a more in-depth investigations involving a greater number of bRo5 compounds and a careful analysis of IMHB patterns is probably needed before introducing congruent states in terms of physicochemical properties. We change the text accordingly.
(3) The procedure for obtaining “energy weighted” ensemble should be described in detail so that it can be reproduced.
We thank the reviewer for pointing this out. We added a paragraph in the Methods
(4) I wonder if molecular dipole moments differ among representative conformers.
The main issue concerning molecular dipole moments regards the accessibility of its calculation. Since no ad hoc method has yet been established for PROTACs, using an adequate level of theory to overcome this issue risks to become too time consuming and beyond the aim of the study. However this is surely an idea that we will pursue in the future.
(5) The term “conformer selection” can be confused with the classical “conformer selection vs induced fit”.
Ok. We rephrased the text to avoid confusion.
Reviewer 2 Report
Dear Authors,
I have reviewed your manuscript, and I am expressing my highly positive feedback. Your study is exciting for the readers of the Pharmaceutics journal, and the obtained results are promising. Since there is some space for improvements, I am requesting minor revisions according to the following comments:
· The space between the word and citations is missing throughout the whole manuscript. Please correct this.
· Please indicate properly which particular Schrödinger Suite was used.
· It is also necessary to precisely mention which program of the Schrödinger suite was used for conformational sampling. I presume MacroModel. Also, it is necessary to cite these programs correctly. Please consult the Schrödinger citation page available at: https://www.schrodinger.com/citations
· It appears that you have used the MOPAC program outside the Schrödinger suite. If that is the case, which tool was used for generating input files for MOPAC16 and for postprocessing?
· If you tend to use the semiempirical methods outside the Schrödinger suite, I recommend visiting the atomistica.online platform available at https://atomistica.online website. You can find out more about this project in the paper: https://doi.org/10.1080/08927022.2022.2126865
This platform is free to use, and you can perform calculations based on PM7 and DFTB methods. Everything is online, the actual calculations are performed on a remote server, and you can finish everything directly from your web browser. You can use both MOPAC and xTB programs. In this way, for some of your future papers, you can maybe try optimizing your molecules with, for example, the GFN2-xTB method, which is a newer approach. In your case, it might be interesting to test the performance of PM7 vs. GFN2-xTB methods.
· Line 244, “OPLSe” should be “OPLS3e”
· Lines 314, 315, “MOPAC2006” should be “MOPAC2016”
Once you address all of the above-mentioned comments, I will gladly review your manuscript again.
Best regards
Author Response
- The space between the word and citations is missing throughout the whole manuscript. Please correct this.
True, we corrected it.
- Please indicate properly which particular Schrödinger Suite was used. It is also necessary to precisely mention which program of the Schrödinger suite was used for conformational sampling. I presume MacroModel. Also, it is necessary to cite these programs correctly. Please consult the Schrödinger citation page available at: https://www.schrodinger.com/citations
Good point, we added the requested information in the Methods section.
- It appears that you have used the MOPAC program outside the Schrödinger suite. If that is the case, which tool was used for generating input files for MOPAC16 and for postprocessing?
We completed the Methods section with the requested information.
- If you tend to use the semiempirical methods outside the Schrödinger suite, I recommend visiting the atomistica.online platform available at https://atomistica.online website. You can find out more about this project in the paper: https://doi.org/10.1080/08927022.2022.2126865.
This platform is free to use, and you can perform calculations based on PM7 and DFTB methods. Everything is online, the actual calculations are performed on a remote server, and you can finish everything directly from your web browser. You can use both MOPAC and xTB programs. In this way, for some of your future papers, you can maybe try optimizing your molecules with, for example, the GFN2-xTB method, which is a newer approach. In your case, it might be interesting to test the performance of PM7 vs. GFN2-xTB methods
We thank the reviewer for the interesting suggestion. Unfortunately, the 100-atom limit for the calculation excludes the application to the majority of PROTACs, including PROTAC-1. For projects involving smaller molecules, we will surely consider this resource.
- Line 244, “OPLSe” should be “OPLS3e”
Ok, done.
- Lines 314, 315, “MOPAC2006” should be “MOPAC2016”
Ok, done